# CHAINGAN: A SEQUENTIAL APPROACH TO GANS

## ABSTRACT

We propose a new architecture and training methodology for generative adversarial networks. Current approaches attempt to learn the transformation from a noise sample to a generated data sample in one shot. Our proposed generator architecture, called *ChainGAN*, uses a two-step process. It first attempts to transform a noise vector into a crude sample, similar to a traditional generator. Next, a chain of networks, called *editors*, attempt to sequentially enhance this sample. We train each of these units independently, instead of with end-to-end backpropagation on the entire chain. Our model is robust, efficient, and flexible as we can apply it to various network architectures. We provide rationale for our choices and experimentally evaluate our model, achieving competitive results on several datasets.

## 1 INTRODUCTION

Generative Adversarial Networks (GANs) are a class of generative models that have shown promising results in multiple domains, including image generation, text generation, and style-transfer (Goodfellow et al., 2014; Zhang et al., 2017; Zhu et al., 2017). GANs use a game-theoretic framework pitting two agents – a discriminator and a generator – against one another to learn a data distribution $P_d$. The generator attempts to transform a noise vector $\boldsymbol{z} \sim \mathcal{N}$ into a sample $\tilde{\boldsymbol{x}}$ that resembles the training data. Meanwhile, the discriminator acts as a classifier trying to distinguish between real samples, $\boldsymbol{x} \sim P_d$, and generated ones, $\tilde{\boldsymbol{x}} \sim P_g$. Equation 1 is the original minimax objective proposed by Goodfellow et al. (2014), which can also be viewed as minimizing the Jensen-Shannon divergence between two distributions, $P_d$ and $P_g$. These agents are most often implemented as neural networks and Goodfellow et al. (2014) proved that, under suitable conditions, the generator distribution $P_g$ will match the data distribution, $P_d$.

$$\min_G \max_D \mathbb{E}_{\boldsymbol{x} \sim P_d}[\log(D(\boldsymbol{x})] + \mathbb{E}_{\boldsymbol{z} \sim \mathcal{N}}[1 - \log(D(G(\boldsymbol{z}))] \tag{1}$$

Despite their success, GANs are notoriously hard to train and many variants have been proposed to tackle shortcomings. These shortcomings include mode collapse (i.e., the tendency for the generator to produce samples lying on a few modes instead of over the whole data space), training instability, and convergence properties. In most of these modified GANs, however, the generator is realized as a single network that transforms noise into a generated data sample.

In this paper, we propose a sequential generator architecture composed of a number of networks. In the first step, a traditional generator (in this paper called the *base generator*) is used to transform a noise sample $\boldsymbol{z}$ into a generated sample $\tilde{\boldsymbol{x_0}}$. The sample is then fed into a network (called *editor*) which attempts to enhance its quality, as judged by the discriminator. We have a number of these *editor* networks connected in a chain, such that the output of one is used as the input of the next. Together, the *base generator* and the *editors* ahead of it are called the *chain generator*. Any preexisting generator architecture can be used for the *base generator*, making it quite flexible. For the *editors*, architectures designed for sample enhancement work well. The whole *chain generator* network can be quite small; in fact, it is smaller than most existing generator networks, yet produces equivalent or better results (Section 4.2).

Each network in the *chain generator* is trained independently based on scores they receive from the discriminator. This is done instead of an end-to-end backpropagation through the whole *base generator + editor* chain (Section 3.3). This allows each network in the chain to get direct feedback on its output. A similar approach has been quite successful in classification tasks. Huang et al.

(2017) showed that in a classification architecture composed of a chain of ResNet blocks, training each of these blocks independently, produces better results than an end-to-end training scheme (He et al., 2015).

In this paper, we propose a sequential generator architecture and a corresponding training regime for GANs. The benefits of this approach include:

- Better performance on some datasets when compared with relevant models (Table 1)
- Efficient in memory and in the number of parameters (Section 7.1).
- *Base generator* itself performs better when trained alongside *editors* (Figure 4a).
- It is flexible and can be adapted to existing network architectures

## 2 BACKGROUND AND RELEVANT WORK

### 2.1 GAN OBJECTIVES

The formulation by Goodfellow et al. (2014) (Equation 1) is equivalent to minimizing the Jensen-Shannon divergence between the data distribution $P_d$ and the generated distribution $P_g$. However, Arjovsky et al. (2017) showed that, in practice, this leads to a loss that is potentially not continuous, causing training difficulty. To address this, they presented the Wasserstein GAN (WGAN) which minimized the Wasserstein (or 'Earth mover') distance between two distributions. This is a weaker distance than Jensen-Shannon and Arjovsky et al. (2017) showed that, under mild conditions, it resulted in a function that was continuous everywhere and differentiable almost everywhere. The WGAN objective can be expressed as:

$$\min_G \max_{D \in \mathcal{D}} \mathbb{E}_{\boldsymbol{x} \sim P_d}[D(\boldsymbol{x})] - \mathbb{E}_{\tilde{\boldsymbol{x}} \sim P_g}[D(\tilde{\boldsymbol{x}})] \tag{2}$$

where $\mathcal{D}$ is the set of all 1-Lipchitz functions. $D$ no longer discerns between real and fake samples but is used to approximate the Wasserstein distance between the two distributions (it is here called a 'critic'). $G$ in turn looks to minimize this approximated distance. Several approaches have been proposed to enforce the Lipchitz constraint on the critic, the most notable being gradient penalty (Gulrajani et al., 2017). This method penalizes the norm of the critic's gradient with respect to its input and was shown to have stable training behavior on various architectures. Due to its strong theoretical and empirical results, we use a modified version of this objective for our work. The objective for WGAN with gradient penalty can be expressed as:

$$L = \mathbb{E}_{\boldsymbol{x} \sim P_d}[D(\boldsymbol{x})] - \mathbb{E}_{\tilde{\boldsymbol{x}} \sim P_g}[D(\tilde{\boldsymbol{x}})] + \lambda \mathbb{E}_{\hat{\boldsymbol{x}} \sim P_{\hat{\boldsymbol{x}}}}[(||\nabla_{\hat{\boldsymbol{x}}} D(\hat{\boldsymbol{x}})||_2 - 1)^2], \tag{3}$$

where $P_{\hat{x}}$ samples uniformly along straight lines between pair of points sampled from the data distribution $P_d$ and the generator distribution $P_g$ Gulrajani et al. (2017). $\lambda$ is the gradient penalty coefficient.

### 2.2 GAN ARCHITECTURES

There have been a diverse range of architectures proposed for the generator and discriminator networks. DCGAN has shown promising results by using a deep convolutional architecture for both the generator and discriminator (Radford et al., 2015). However, this uses a single large generator network instead of the sequential architecture that we have developed. Our architecture provides similar results as DCGAN while being more efficient, both in network parameters (Section 4.2) and in memory. EnhanceGAN, proposed by Deng et al. (2017), successfully use GANs for unsupervised image enhancement. In this model, the generator is given an image, and it tries to improve its aesthetic quality. The *editor* part of our network is similar to this as they are responsible for enhancing images in an unsupervised fashion.

Sequential generator architectures have been proposed for tasks where the underlying data has some inherent sequential structure. TextGAN uses an LSTM architecture for the generator to generate a

sentence, one word at a time (Zhang et al., 2017). Ghosh et al. (2016) used an RNN-style discriminator and generator to solve abstract reasoning problems. This attempts to predict the next logical diagram, given a sequence of past diagrams. C-RNN-GAN is another example that uses GANs on continuous sequential data like music (Mogren, 2016). Unlike these works, we do not assume the data is inherently sequential nor do we produce a single sample by generating a number of smaller units; rather, the output of any network in the *chain generator* is a whole sample, and the succeeding ones try to enhance it. Moreover, each *editor* network has its own set of parameters, unlike LSTMs or RNNs where these are shared. This allows each *editor* to be independent and makes our model efficient as we do not have to backpropagate across time.

The concept of using multiple generators has been proposed previously. Hoang et al. (2017) used multiple generators which share all but the last layer, trained alongside a single discriminator and a classifier. Their work has shown resistance to mode collapse and achieves state-of-the-art results on some datasets. Their work was influenced by MIX+GAN, which showed strong theoretical and empirical results by using a mixture of GANs instead of a single generator and discriminator (Arora et al., 2017). Notably, they showed that using a mixture of GANs guarantees existence of approximate equilibrium, leading to stable training. However, a drawback of their work was the large memory cost of using multiple GANs - they suggest a mixture size of 5. Our work is similar to these in that we use multiple generators. The output of each *editor* is a different transformation on the noise vector $z$ and can be viewed as originating from different generators. However, the generators in our model are not independent; rather, they are connected in a chain and meant to build upon the previous ones.

## 3 MODEL

### 3.1 MOTIVATION

A clear advantage of a sequential approach is that each output in the sequence is conditioned on the previous one, simplifying the problem of generating a complicated sample by mapping them to a sequence of simpler problems. A sequential approach also means that loss or other latent information in the intermediate steps can be used to guide the training process, which is not possible in one-shot generation.

As such, we propose a sequential generator architecture and an associated training objective for GANs. We observe that the challenge of generating a data sample from noise can be divided into two components: generating a crude representation of what that sample might be, and then iteratively refining it (Figure 1). A generator trained on images may first generate a rough picture of a car or a boat, e.g., but it can then be successively edited and enhanced. Intuitively, this is akin to the editorial process wherein a writer produces a rough sketch of an article, which then passes through a number of editors before the finished copy is published.

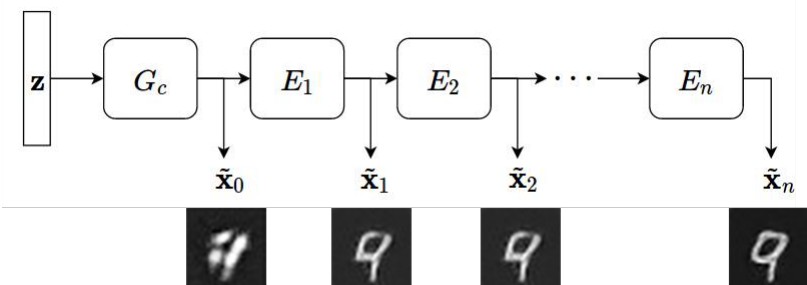

Figure 1: *Chain generator* model, with noise $\mathbf{z}$, *base generator* $G_b$, and *editors*, $E_i$.

### 3.2 ARCHITECTURE

We use a WGAN with gradient penalty as the starting point for our architecture due to its theoretical robustness and training stability (Section 2.1). However, instead of using a single generator, we use

the *chain generator* consisting of a *base generator*, denoted by $G_b$ and several *editors*, denoted by $E_i$. The *base generator* takes in $z \sim \mathcal{N}$, and produces a sample $\tilde{x}_0$ attempting to fool the critic, as in the traditional GAN architecture. The *editors* are connected in a chain where the $i^{th}$ one takes as input a sample $\tilde{x}_{(i-1)}$ and tries to produce an enhanced version as the output, $\hat{x}_i$ (Figure 1). Thus, the output of each *editor* is conditioned on the previous one. Our network can be expressed recursively as:

$$\tilde{x}_0 = G_b(z)$$
$$\tilde{x}_i = E_i(\tilde{x}_{(i-1)}) \tag{4}$$

Each of the intermediate samples in the chain, $\tilde{x}_i$, is an attempt to fool the critic. In the WGAN formulation, the critic is used to approximate the Wasserstein distance between the data distribution $P_d$ and the generated distribution $P_g$, where $P_g$ is obtained by transforming $z \sim \mathcal{N}$ using the network $G$. In our formulation, each output $\tilde{x}_i$ is obtained by the following transformation on $z$: $E_i(E_{i-1}(\ldots E_1(G_b(z)))))$. As such, $x_i$ can be viewed as being sampled from distribution $P_{gi}$ and a critic is needed to approximate the Wasserstein distance between $P_d$ and $P_{g_i}$ (see Section 7 for a detailed explanation). For a chain of $n$ *editors* plus the *base generator*, $n+1$ critics are needed. For the $i^{th}$ *editor* and critic, the objective can be expressed as:

$$L_i = \mathop{\mathbb{E}}_{x \sim P_d}[D_i(x)] - \mathop{\mathbb{E}}_{\tilde{x}_i \sim P_{g_i}}[D_i(\tilde{x}_i)] + \lambda \mathop{\mathbb{E}}_{\hat{x} \sim P_{\hat{x}_i}}[(||\nabla_{\hat{x}_i} D_i(\hat{x})||_2 - 1)^2] \tag{5}$$

## 3.3 TRAINING

Each network in the *chain generator*, whether the *base generator* or any of the *editors*, is trained independently, based on critic scores for that network's output (Section 7.1). At every iteration, we randomly choose one of them to update, and use their output to also train the corresponding critic, $D_i$. This means that the goal for any network in the *chain generator* is to ensure its outputs are as realistic as possible. Without loss of generality, we can express the training rule for the *chain generator* using canonical gradient descent with learning rate $\alpha$ as:

$$\theta_{G_b}^{t+1} = \theta_{G_b}^t - \alpha \nabla_{\theta_{G_b}} D_0(\tilde{x}_0)$$
$$\theta_{E_i}^{t+1} = \theta_{E_i}^t - \alpha \nabla_{\theta_{E_i}} D_i(\tilde{x}_i) \tag{6}$$

Our training regime is in contrast to performing end-to-end backpropagation (from the last *editor* all the way to the *base generator*). The motivation for this is three-fold. First, we want each *editor* to do its best in generating realistic samples and receive direct feedback on its performance. This way, we can utilize the intermediate samples generated in the chain to guide the training process. So *editor i* can start with the distribution $P_{g_{i-1}}$ and use the critic feedback to construct a $P_{g_i}$ that is closer to $P_d$. Another benefit of this is that we can possibly use fewer *editors* during evaluation. For example, we can train with the *chain generator* composed of $G_b, E_1, E_2, \ldots, E_n$, but in evaluation observe that after *editor* $k$ ($k \leq n$), the quality of the samples saturate or do not improve significantly (Figure 2). As such, during evaluation we can cutoff the *chain generator* at *editor* $k$, and use this smaller chain for downstream tasks, making it more compact. Performing end-to-end backpropagation would mean that only the samples generated by the last *editor* would be viewed by the critic, and the goal of the *chain generator* would be to generate good samples by the last *editor*. That network could no longer be made compact at evaluation, and the number of *editors* to use would be an additional hyper-parameter.

The last rationale behind our training approach is efficiency. Observe that in the canonical GAN architecture, the gradient computation when updating the generator is $\frac{\partial}{\partial \theta_G} D(G(z))$, which involves traversing both the generator and critic graphs, which can be expensive for large generator networks. Performing end-to-end backpropagation on our *chain generator* would cause a similar issue as the entire chain would need to be retained alongside the critic. However, training each of these networks separately means that the gradient computation for an update step becomes $\frac{\partial}{\partial \theta_{E_i}} D_i(E_i(\tilde{x}_{i-1}))$, so only a single *editor* or *base generator* and its critic need to be traversed and loaded on the GPU (Section 7.1). Since these networks are quite small, our model can be made very efficient.

$$G_b \qquad E_1 \qquad E_2 \qquad E_3 \qquad E_4 \qquad E_5$$

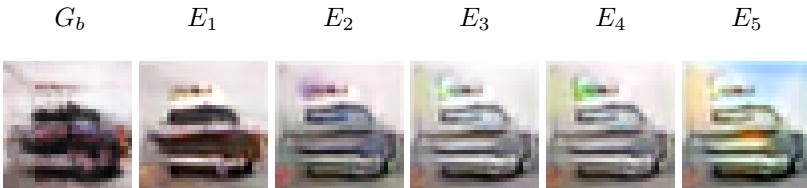

Figure 2: In this example, the first few *editors* significantly enhance the image of a car, but after Editor 3, they tend to more subtle.

We abide by the recommendations of Arjovsky et al. (2017) to train the critic more than the generator in each iteration and use a gradient penalty to enforce the Lipchitz constraint. We use the Adam optimizer (Kingma & Ba, 2014). Algorithm 1 explains our training process.

---

**Algorithm 1:** ChainGAN Training
Our default values are $\lambda = 10$, $n_{critic} = 5$, $\alpha = 0.0001$, $(\beta_1, \beta_2) = (0.5, 0.9)$, batch size = 64.

---

**Require:** initial critic parameters $w^0$ and initial *chain generator* parameters $\{\theta_0^0 \ldots \theta_n^0\}$, where
$\qquad$ $\theta_i$ are the parameters for the $i^{th}$ network in the *chain generator*.
**Require:** gradient penalty coefficient $\lambda$, batch size $m$, critic iterations per generator iteration
$\qquad$ $n_{critic}$, and Adam hyperparametes $\alpha$, $\beta_1$, $\beta_2$,
**while** $\theta_0, \ldots \theta_n$ has not converged **do**
$\quad$ $i \sim \text{Cat}(0, n)$ ;
$\quad$ **for** $j = 1, \ldots, n_{critic}$ **do**
$\quad\quad$ **for** $k = 1, \ldots, m$ **do**
$\quad\quad\quad$ Sample real data, $\boldsymbol{x} \sim P_d$, and noise, $\boldsymbol{z} \sim \mathcal{N}(0, 1)$;
$\quad\quad\quad$ Sample $\boldsymbol{\epsilon} \sim \mathcal{U}(0, 1)$;
$\quad\quad\quad$ $\tilde{\boldsymbol{x_i}} \leftarrow \text{E}_i(E_{i-1}(\ldots G_b(\boldsymbol{z})))$;
$\quad\quad\quad$ $\hat{\boldsymbol{x}} \leftarrow \epsilon \boldsymbol{x} + (1 - \epsilon)\tilde{\boldsymbol{x_i}}$;
$\quad\quad\quad$ $L^{(k)} \leftarrow D_i(\boldsymbol{x}) - D_i(\tilde{\boldsymbol{x}}) + \lambda(||\nabla_{\hat{\boldsymbol{x}}} D_i(\hat{\boldsymbol{x}})||_2 - 1)^2$;
$\quad\quad$ **end for**
$\quad\quad$ $w^{t+1} \leftarrow \text{Adam}(\nabla_{w^t} \frac{1}{m} \sum_{k=1}^m L^{(k)}, w^t, \alpha, \beta_1, \beta_2)$;
$\quad$ **end for**
$\quad$ Sample a batch of $\{\boldsymbol{z}\}_{k=1}^m \sim \mathcal{N}$;
$\quad$ $\{\tilde{\boldsymbol{x_i}}\}_{k=1}^m \leftarrow \{E_i(E_{i-1}(\ldots G_b(\boldsymbol{z_k})))\}_{k=1}^m$;
$\quad$ $\theta_i^{t+1} \leftarrow \text{Adam}(\nabla_{\theta_i^t} \frac{1}{m} \sum_{k=1}^m -D_i(\tilde{\boldsymbol{x_i}}), \theta_i^t, \alpha, \beta_1, \beta_2)$;
**end while**

---

## 4 EXPERIMENTS

We deploy our model on several datasets that are commonly used to evaluate GANs, including: (i) MNIST – 28x28 gray-scale images of hand-written digits (LeCun & Cortes, 2010); (ii) CIFAR10 – 32x32 colour images across 10 classes (Krizhevsky et al.); (iii) CelebA face dataset – 214x110 colour images of celebrity faces (Liu et al., 2015). For the CIFAR10 dataset, we compute the inception score and compare it against several GAN variants (Salimans et al., 2016). Our goal is to show that our model can be used with existing generator architectures to provide competitive scores while using a smaller network.

We implement multiple critics by sharing all but the last layer (Section 7.3). To verify our results were not due to this, we also experiment with using a single critic, which did not degrade results. Each *editor* is composed of a few residual blocks since Deng et al. (2017) showed that such an architecture performs well for unsupervised image enhancement tasks (see Section 7.4 for details). We use this *editor* architecture for all our experiments and have five such networks connected in a chain. A key goal when designing our networks was efficiency; in fact, all implementations of our *chain generator* have fewer parameters than comparable models.

## 4.1 MNIST

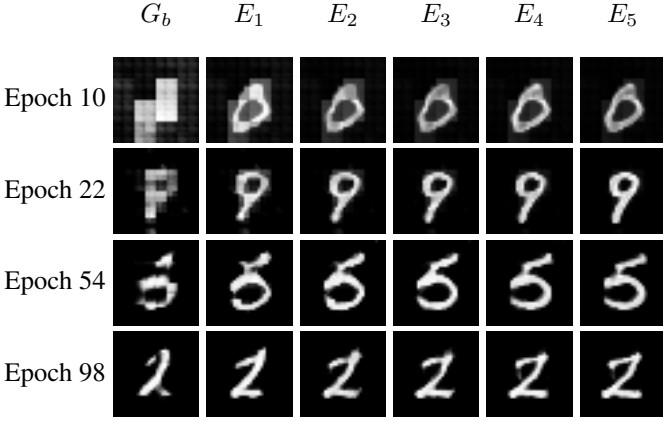

Figure 3: Randomly selected MNIST examples from our DCGAN with editors model at different epochs.

Our *base generator* and the critics use a convolutional architecture similar to DCGAN. The entire *chain generator* network is quite small, using approximately a third of the number of parameters as the DCGAN generator. Early in the training, the *base generator's* samples are crude, but the *editors* work to enhance it significantly (Figure 3). As the training progresses, the *base generator's* samples increase in quality and the *editor's* effect becomes more subtle. Visually, it appears that the *editors* collaborate and build upon each other; the class and overall structure of the image remain the same throughout the edit-process.

## 4.2 CIFAR10

Table 1: Inception scores on unlabelled CIFAR with and without the *chain generator* approach trained with WGAN+GP. $d$ refers to the depth of the first conv layer in DCGAN architectures. See section 7.4 for details about architecture.

| Model | Generator Arch. | $G_b$ Score | Best $E_i$ Score | ∼Params |
|---|---|---|---|---|
| 1 | Small DCGAN ($d$=256) | 5.02±0.05 | N/A | 700,000 |
| 2 | **Tiny DCGAN ($d$=128) + Editors** | **5.00±0.09** | **5.67±0.07** (Edit 3) | **866,000** |
| 3 | **Small DCGAN + Editors (one Critic)** | **5.24±0.04** | **6.05±0.08** (Edit 2) | **1,250,000** |
| 4 | **Small DCGAN + Editors (multi Critic)** | **5.63±0.06** | **5.86±0.05** (Edit 2) | **1,250,000** |
| 5 | Small DCGAN + Editors (end-to-end) | N/A | 2.56±0.02 | 1,250,000 |
| 6 | DCGAN - WGAN+GP ($d$=512) | 5.18±0.05 | N/A | 1,730,000 |
| 7 | Small ResNet | 5.75±0.05 | | 720,000 |
| 8 | **Small ResNet + Editors** | **6.35±0.09** | **6.70±0.03** (Edit 3) | **1,000,000** |
| 9 | ResNet - WGAN+GP | 6.86±0.04 | N/A | 1,220,000 |

We experiment with different architectures for the *base generator*. These include variants of DC-GAN and a ResNet architecture proposed by Gulrajani et al. (2017). We add a chain of five *editors* to these *base generators* and train them according to Algorithm 1. We compare them against the original (and larger) versions of these architectures. For original architectures, we use the tuned hyper-parameters recommended in the pertinent paper/implementation. For the *ChainGAN* variants, we use default parameters instead. We wanted to evaluate the robustness of our model and believe that better results than ours can be obtained by tuning hyper-parameters. All models compared are trained to the same number of epochs using the WGAN formulation with a gradient penalty term (WGAN+GP). Table 1 contains the results of these experiments.

Based on inception scores, our DCGAN variant of the *chain generator* using multiple critics (model 2 and 4) was able to outperform the base DCGAN model (model 6) while using fewer parameters. In a related experiment, we train the *base generator* by itself (i.e., without *editors*) against the critic (model 1), to note the effect of the *editors* on the *base generator*. In this case, the *base generator* exhibits better inception scores when trained alongside *editors* than without them, as shown in Fig. 4a. This suggests that the sequential model not only improves the overall sample quality, but may also improve *base generator* training. To discern the role of our training regime, we run an experiment that uses end-to-end backpropagation on a *chain generator* with 5 *editors* (model 5), noting a significant drop in performance.

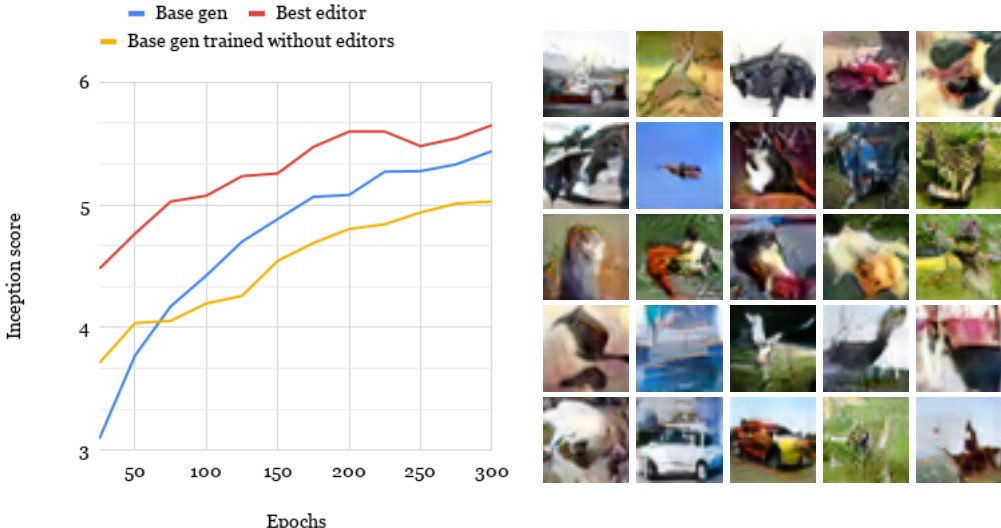

Figure 4: (a) Inception scores across epochs of model 4 vs model 1 (b) Randomly selected CIFAR-10 examples from editor 3 of Model 8.

Table 2: Comparison of inception scores

| Method (unlabelled) | Inception scores |
| --- | --- |
| Real Data | 11.24±0.12 |
| WGAN (Arjovsky et al., 2017) | 3.82±0.06 |
| MIX + WGAN (Arora et al., 2017) | 4.04±0.07 |
| DCGAN - WGAN+GP * | 5.18±0.05 |
| ALI (Dumoulin et al., 2016) | 5.35±0.05 |
| BEGAN (Berthelot et al., 2017) | 5.62 |
| **Small DCGAN (WGAN+GP) + Editors**[*] | 6.05±0.04 |
| **Small ResNet (WGAN+GP) + Editors**[*] | 6.70±0.02 |
| ResNet - WGAN+GP * | 6.86±0.04 |
| EGAN-Ent-VI (Dai et al., 2017) | 7.07±0.10 |
| MGAN (Hoang et al., 2017) | 8.33±0.10 |

The variant of *ChainGAN* using a ResNet *base generator* also uses a smaller network (model 8), while achieving comparable scores. While performing slightly worse than base on the standard inception score (IV2-TF), we outperform the base model in the IV3-Torch inception model used in Barratt & Sharma (2018). We also see the same effect as above: results from the *base generator* are better when in chain and trained alongside *editors* (model 8) rather than being trained by itself (model 7). Figure 4b shows random samples from Editor 3 of the Small ResNet + Editors model.

Table 2 compares our models against different GAN variants. Entries denoted by $*$ refer to models that we implemented (all using pyTorch). The ResNet - WGAN+GP is the model proposed by Gulrajani et al. (2017) and we closely followed all their recommendations yet were unable to achieve their stated inception score of 7.86.

## 4.3 CELEBA

We also trained our model on the celebrity face dataset (Liu et al., 2015). We do not center-crop the images, which would focus only on the face; rather, we downsample but maintain the whole image. As in the previous experiments, the *editors* build upon one another rather than working at cross purposes. Section 7.2 shows some samples generated by our model.

## 5 DISCUSSION AND FUTURE WORK

In our experiments, we were able to successfully train a sequential GAN architecture to achieve competitive results. Our model uses a smaller network and the training regime is also more efficient than the alternatives (Section 7.1). We trained with 5 *editors*, but in evaluation needed $k \leq 5$ *editors* to achieve the best scores; this suggests that we can use a smaller network during evaluation or for downstream tasks. Our results yielded diverse samples and we do not notice any visual signs of mode collapse (Figure 4b). To ensure that our generator architecture and training scheme are responsible for the performance, we run experiments with a single critic and see that results do not degrade (see model 3 in Table 1); but with end-to-end backpropagation through the *chain generator* results degraded significantly (see model 5 in Table 1).

We also experimented with inducing a game between the *editors* themselves, using their loss functions. In one formulation, $E_i$'s loss was the difference of the critic scores between itself and the previous *editor* scaled by $\lambda$, $L_i = D_i(\tilde{\boldsymbol{x_i}}) - \lambda D_{i-1}(\tilde{\boldsymbol{x_{i-1}}})$. As such, each *editor* would compete against the previous one and try to outperform it. In another formulation, each *editor*'s loss was a discounted (by $\lambda$) sum of all the critic scores ahead of it, $L_i = D_i(\tilde{\boldsymbol{x_i}}) + \lambda D_{i+1}(\tilde{\boldsymbol{x_{i+1}}}) + ... + \lambda^{n-i} D_n(\tilde{\boldsymbol{x_n}})$. This would force earlier *editors* to take on more responsibility since they influence those further ahead. Although promising, these methods often led to unstable training. Future work could focus on training stability with these approaches as there is merit in exploring strategies between *editors*.

### 5.1 OPEN CHALLENGES

The idea of splitting sample generation into a multi-step process is very flexible and can be extended in a number of ways. Currently, *editors* are completely unsupervised, but this can be extended to make each *editor* responsible for a different feature. One approach might build upon InfoGAN, which provides the generator with several latent variables, along with noise, and tries to maximize the mutual information between the generated samples and the latent variables (Chen et al., 2016). Using a *ChainGAN* approach, we could use the *base generator* to simply generate a sample, and each *editor* would be responsible for a different latent variable, with a goal of maximizing mutual information between that variable and its output.

*ChainGAN* may also prove effective in text generation – a domain with which GANs struggle. In this context, the *base generator* could be responsible for generating a bag of words, and the *editors* would be responsible for reordering them, e.g., for coherence. *Editors* could similarly be used change the tone or sentiment of text appropriately.

## 6 CONCLUSION

In this paper, we present a new sequential approach to GANs. Instead of attempting to generate a sample from noise in one shot, our model first generates a crude sample via the *base generator* and then successively enhances it through a chain of networks called *editors*. We use multiple critics, each corresponding to a different network in the *chain generator*. Our model is efficient and we successfully trained on a number of datasets to achieve competitive results, outperforming some existing models. Furthermore, our scheme is very flexible and there are several avenues for extension.

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

# 7 APPENDIX

## 7.1 TRADITIONAL GAN TRAINING VS CHAINGAN TRAINING

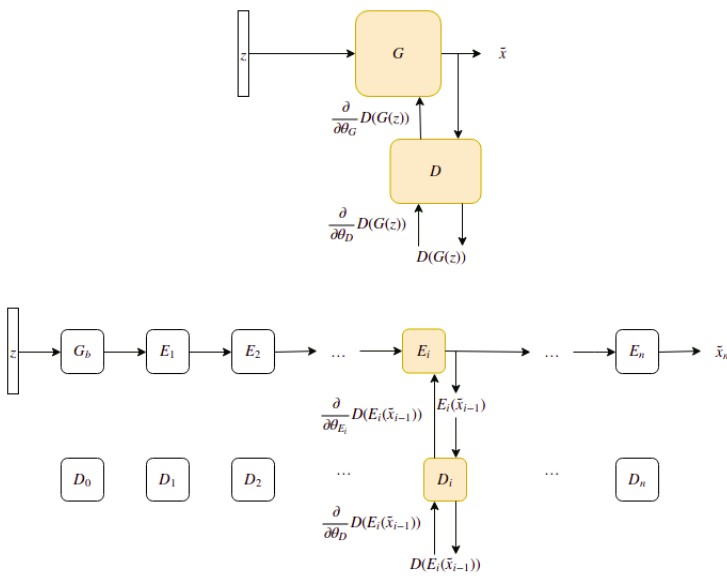

Figure 5: Whereas backpropagation for traditional GANs (top) require retaining the entire generator graph, ChainGAN requires the graph of one randomly selected *editor* per iteration (bottom), which has a fraction of the number of parameters. Orange boxes represent the parameters involved in the backpropagation of loss gradients.

## 7.2 CELEBA RESULTS

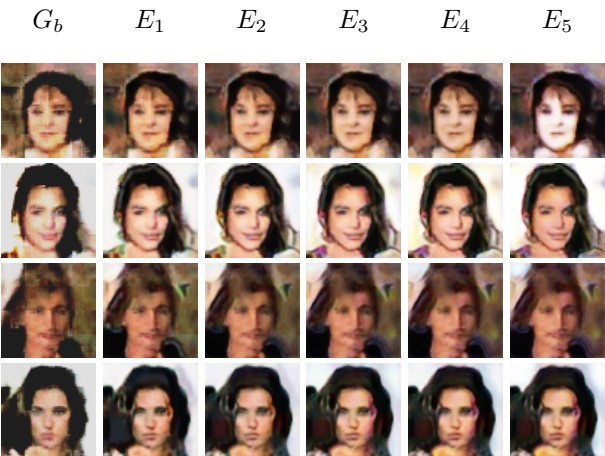

Figure 6: CelebA samples. *Editors* progressively smooth irregularities of the initial generated images.

### 7.3 Rationale for multiple critics

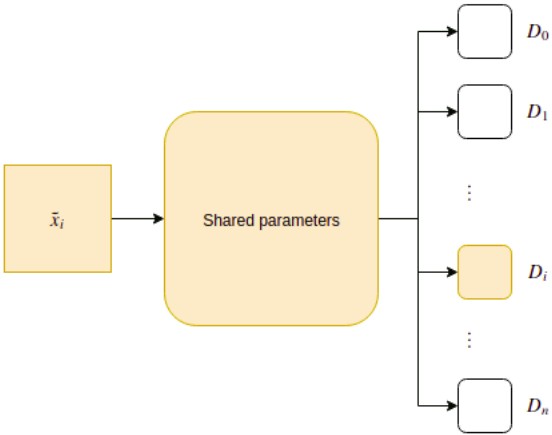

Figure 7: Multiple critics share convolution parameters and differ only in the last linear layer. Critic $D_0$ is responsible for evaluating *base generator $G_b$*'s output $\tilde{x}_0$, $D_i$ is responsible for evaluating *editor $E_i$*'s output $\tilde{x}_i$, and so on.

The Wasserstein distance can be intuitively interpreted as the cost of the cheapest transport strategy that transforms one distribution to another. Between two distributions $P_g$ and $P_d$, this can be expressed as:

$$W(P_g, P_d) = \inf_{\gamma \epsilon \prod(P_d, P_g)} \mathbb{E}_{(x,y)\sim\gamma}[||x - y||] \tag{7}$$

where $\prod(P_d, P_g)$ is the set of all distributions whose marginals are $P_g$ and $P_d$. Arjovsky et al. (2017) show that using the Kantorovich-Rubenstein duality and a slight reframing, this objective can be expressed as:

$$W(P_g, P_d) = \max_{w\epsilon\mathcal{W}} \mathbb{E}_{\boldsymbol{x}\sim P_d}[f_w(x)] - \mathbb{E}_{\boldsymbol{z}\sim\mathcal{N}}[f_w(G(\boldsymbol{z}))] \tag{8}$$

Here, $\{f_w\}_{w\epsilon\mathcal{W}}$ is the set of all $K$-Lipchitz functions for some $K$. We can use a neural network $D$ to approximate the function $f_w$. Now that we have an approximation for the Wasserstein distance between $P_d$ and $P_g$, the generator network seeks to minimize this, giving us the WGAN formulation of:

$$\min_G \max_{D\in\mathcal{D}} \mathbb{E}_{\boldsymbol{x}\sim P_d}[D(\boldsymbol{x})] - \mathbb{E}_{\tilde{\boldsymbol{x}}\sim P_g}[D(\tilde{\boldsymbol{x}})] \tag{9}$$

In our *ChainGAN* model, the output of each *editor* can be seen as a different transformation of the noise vector $\boldsymbol{z} \sim \mathcal{N}$. For example, *Editor i*'s output $\tilde{\boldsymbol{x}_i}$ is obtained by the following transformation on $\boldsymbol{z}$: $E_i(E_{i-1}(\ldots E_1(G_b(\boldsymbol{z}))))$. Thus, to approximate the Wasserstein distance between $P_d$ and $P_{g_i}$, we use a network $D_i$. For a *base generator* and *n editors*, we have *n+1* critics.

## 7.4 ARCHITECTURES

### 7.4.1 RESNET ARCHITECTURES

The results before each ResBlock (of any type) were added to the output of that ResBlock after undergoing a resample (of the same type as the ResBlock in question) and a dimension invariant convolution with filter size $[1 \times 1]$. The architecture choices for the different ResBlocks all come from the WGAN-GP paper (Gulrajani et al., 2017). ResBlock type I is composed of a convolution with padding 1 followed by a ReLU activation function followed by another similar convolution followed mean pooling. ResBlock type II is composed of a ReLU activation function followed by a convolution with padding 1, then another ReLU followed by a similar convolution again and a mean pooling layer at the end. ResBlock type III is the same as ResBlock type II except it includes a batch normalization layer before each ReLU and has an upsampling layer instead of the mean pooling layer. ResBlock type IV is similar to types II and III except it has no resampling layer. The last convolution layer is just a convolution with padding 1.

| **Critic** $D$ ($d = 128$) | | | |
|---|---|---|---|
| | Kernel Size | Resample | Output shape |
| Input Image | - | - | $3 \times 32 \times 32$ |
| Residual Block Type I | $[3 \times 3]$ | Down | $d \times 16 \times 16$ |
| Residual Block Type II | $[3 \times 3]$ | Down | $d \times 8 \times 8$ |
| Residual Block Type II | $[3 \times 3]$ | Down | $d \times 4 \times 4$ |
| Residual Block Type II | $[3 \times 3]$ | Down | $d \times 4 \times 4$ |
| Linear ($n$ such layers) | - | - | 1 |

| **Base Generator** $G_b$ ($d = 96$ or $d = 128$) | | | |
|---|---|---|---|
| | Kernel Size | Resample | Output shape |
| $z$ | - | - | 128 |
| Linear | - | - | $d \times 4 \times 4$ |
| Residual Block Type III | $[3 \times 3]$ | Up | $d \times 8 \times 8$ |
| Residual Block Type III | $[3 \times 3]$ | Up | $d \times 16 \times 16$ |
| Residual Block Type III | $[3 \times 3]$ | Up | $d \times 32 \times 32$ |
| Convolution | $[3 \times 3]$ | - | $3 \times 32 \times 32$ |

| **Editor** $E$ ($d = 34$) | | | |
|---|---|---|---|
| | Kernel Size | Resample | Output shape |
| Previous Image | - | - | $3 \times 32 \times 32$ |
| Residual Block Type IV | $[3 \times 3]$ | - | $d \times 32 \times 32$ |
| Residual Block Type IV | $[3 \times 3]$ | - | $d \times 32 \times 32$ |
| Residual Block Type IV | $[3 \times 3]$ | - | $d \times 32 \times 32$ |
| Convolution | $[3 \times 3]$ | - | $3 \times 32 \times 32$ |

### 7.4.2 DCGAN ARCHITECTURE

Each generator convolution transpose refers to a layer with a convolution transpose of stride 2 and no padding, followed by batch normalization, and then a ReLU activation function. Each critic convolution refers to a convolution operation with stride 2 and padding of 1 followed by a LeakyReLU activation function. Notice here that we do not use batch normalization for the critic. The *editor* used with the DCGAN variant shares its architecture with its counterpart in the ResNet section and is only reproduced here for clarity.

**Critic** $D$ ($d = 512$)

|  | Kernel Size | Output shape |
|---|---|---|
| Input Image | - | $3 \times 32 \times 32$ |
| Convolution | - | $d \times 4 \times 4$ |
| Convolution | $[3 \times 3]$ | $d \times 16 \times 16$ |
| Convolution | $[3 \times 3]$ | $2d \times 8 \times 8$ |
| Convolution | $[3 \times 3]$ | $4d \times 4 \times 4$ |
| Linear ($n$ such layers) | - | 1 |

**Base Generator** $G_b$ ($d = 256$ or $d = 512$)

|  | Kernel Size | Output shape |
|---|---|---|
| $z$ | - | 128 |
| Linear | - | $d \times 4 \times 4$ |
| Convolution Transpose | $[2 \times 2]$ | $\frac{d}{2} \times 8 \times 8$ |
| Convolution Transpose | $[2 \times 2]$ | $\frac{d}{4} \times 16 \times 16$ |
| Convolution Transpose | $[2 \times 2]$ | $3 \times 8 \times 8$ |

**Editor** $E$ ($d = 34$)

|  | Kernel Size | Resample | Output shape |
|---|---|---|---|
| Previous Image | - | - | $3 \times 32 \times 32$ |
| Residual Block Type IV | $[3 \times 3]$ | - | $d \times 32 \times 32$ |
| Residual Block Type IV | $[3 \times 3]$ | - | $d \times 32 \times 32$ |
| Residual Block Type IV | $[3 \times 3]$ | - | $d \times 32 \times 32$ |
| Convolution | $[3 \times 3]$ | - | $3 \times 32 \times 32$ |

## 7.5 ADDITIONAL CIFAR-10 RESULTS

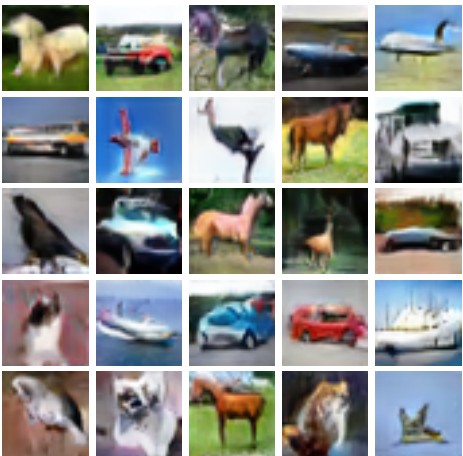

Figure 8: Selected outputs from using ResNets for both the generator and 5 editors.

