# OpenReview forum: "ChainGAN: A sequential approach to GANs"
_ICLR.cc/2019/Conference_

### Official Review · AnonReviewer2 · 2018-10-31
**Interesting direction, not very novel, and with important flaws in the paper.**

**Rating:** 4
**Confidence:** 4

**Review:**

Summary:
The authors present a straightforward method to improve generative quality of GANs that can allow for fewer parameters by separating the task into a basic-generation followed by a chain of multiple edits to the base generation with different editor networks. The fact that separating the task allows for smaller networks is key to the reduction in parameters.  Each editor is trained separately in alternance, with its associated critic. Authors test their approach mostly on CIFAR10, as well as CelebA and MNIST.

Pros:
- The proposed method is simple and makes intuitive sense. Having editor generators acting like highly-conditioned GANs should make their job easier to produce better samples.

- Empirical results show that when removing editors for evaluation, some well-known architecture (DCGAN - WGAN+GP) can be outperformed with less parameters when comparing IS scores.

- Interesting leads and negative results are discussed in Section 5.

Cons :
- Comparisons with end-to-end training seems inadequate. The authors invalidate the end-to-end approach with two arguments; (1) that it doesn’t allow for the removal of superfluous editors once the training is done (Section 3.3), and (2) that IS scores are significantly lower (Section 4.2, Table1). It seems that both these statements are true simply because end-to-end learning is performed with a single score outputted by a single critic at the end of the chain, while it would be entirely possible and simple to keep all discriminators and associated losses and train end-to-end. This would still push all editors to produce good samples, thus allowing removal of editors at test-time, and would probably yield better results than those reported in Table 1. It could also invalidate the results shown in Figure 4 (left). For me this is an important missing comparison, as it might even yield better results than the proposed approach and invalidates one of the proposed advantage of the method.

- I think this idea of sequential generation has been explored before, (e.g. StackGAN [1, 2] or LAPGAN [3] and others?), in which unconditional image generation is performed on relatively complicated datasets with a somewhat more principled way of actually simplifying the task of the base generator. Therefore, I think important citations and valid comparisons are missing.

- The only reported metric is the Inception Score (IS), while most of the recent literature agrees that the Fréchet Inception Distance is a better metric. I think FID should be presented as well to be better aligned with recent literature, even in cases where comparison with previously reported performance is impossible (if previous works only presented IS). If you want to be compared to in future work, I think this is necessary.

- It would be a good addition to have FID/IS scores for each editor output, as we could see the quantitative increase in performance at each editing step.

- In Section 4.2, you specify that all experiments are done using the WGAN-GP training formulation. Looking at Table 1 this is unclear, as you specify this training scheme only for model 6 and model 9. If all models use the same training scheme, this information should be absent from the Table.

- CelebA results. Experiments are reported in the main text, without any results, which are only in the Appendix. These results don’t show any quantitative metrics and are visually disappointing. It is hard to see if the editors actually improve the generation.

- Some of the main results or discussions are based on Section 7, which is not the main article even though it is used as another Section instead of an Appendix. I think Section 7 should be separated into Appendices A, B, etc. Maybe some important aspects of the research presented could fit into the main text, given some removal of repetitions, and some compression of the intro to GANs, which should be vastly known by the ICLR community by now.

- Wrong citation format at the end of Section 2.1.

- Section 3.3 : “train the critic more than the generator in each iteration”. This could be clarified by stating exactly what you do (training the critic for k steps for each generator steps).

- It’s not always clear what the boldface represents throughout tables.

- The discussion in Section 5 about promising techniques explored is somewhat disappointing as efforts to investigate why training failed are not apparent.

- Every result shown from the proposed method is performed with ‘small’ or ‘tiny’ versions of existing architectures. This method could have additional value if it could boost performance on the same architecture, even if there are added editors and trainable parameters. The fact that such results are absent makes me suspicious of such a behavior.

Overall I think this paper presents a relevant and interesting idea. However I think this idea has been explored before with more convincing results and with a more principled approach. There are some important flaws in the comparisons made to assess the advantages of the method, and the overall results fail to convince of any important benefit. Based on the pros/cons stated above, I think this paper does not reach the required quality for acceptance in ICLR.

[1] StackGAN: Text to Photo-realistic Image Synthesis with Stacked Generative Adversarial Networks (Zhang et al. 2017)
[2] StackGAN++: Realistic Image Synthesis with Stacked Generative Adversarial Networks (Zhang et al. 2017)
[3] Deep Generative Image Models using a Laplacian Pyramid of Adversarial Networks (Denton et al. 2015)

---

### Official Review · AnonReviewer1 · 2018-11-02
**interesting idea, insufficiently fleshed out**

**Rating:** 4
**Confidence:** 4

**Review:**

The authors propose `ChainGAN, a GAN architecture where the generator is supplemented with a series of editors that iteratively improve image quality. In practice, the algorithm also uses multiple critics (discriminators), although this is not explained until the Experiments section.

The paper contains the germ of a powerful idea. However, it feels as if the authors haven't yet come to grip with their own idea and architecture. Currently, the role of the editors feels underspecified: it is unclear (and unexplored?) what architectures make for good editors; exactly how editors should interact with the various losses; and what the role of the critics (ideas are proposed in related work) should be. In the experiments, the editors sharpen image quality, but the tradeoffs are not explored. Are more editors always better? When does it saturate? Why? Adding a few editors and critics makes the architecture more parameter-efficient, but increases the number of losses. What happens to wall-clock training time? Moreover, the paper is conflicted about the role of the critic(s). Is the core idea to have multiple generators, discriminators, or both? What is moving the needle?

---

### Official Review · AnonReviewer3 · 2018-11-02

**Rating:** 4
**Confidence:** 4

**Review:**

The paper proposes a GAN variant, called ChainGAN, which expresses the generator as a "base generator" -- which maps the noise vector to a rough model sample -- followed by a sequence of "editors" -- which progressively refine the sample. Each component of the generator is trained independently to fool its own separate discriminator, without backpropagating through the entire chain of editors. The proposed ChainGAN model is trained on MNIST, CIFAR10, and CelebA. The paper presents model samples for all three datasets, as well as Inception scores for CIFAR10.

I find the proposed idea simple and elegant but the evaluation lacking, and as such I’m a bit hesitant to outright recommend accepting the paper:

- Evaluation is not very extensive or detailed. Inception scores are shown only for CIFAR10 and using two base generator architectures. The Inception score has known limitations, and I would have expected the authors to also provide FID scores. The main takeaway is also not articulated very clearly. As far as I can tell it appears to be that ChainGAN allows to achieve similar performance with less tunable parameters, but Table 1 shows mixed results, where ChainGAN outperforms the baseline DCGAN architecture using fewer parameters but underperforms the baseline ResNet architecture.
- The way the experimental section is organized made it difficult for me to find my way around. For example, subsection titles are hard to locate due to the fact that figures and tables were placed immediately underneath them. Overall when the flow of the text is interrupted by a figure, it’s hard to locate where to resume reading.
- There is a connection to be made with other sequential generation approaches (not to be confused with sequence generation) such as LAPGAN, DRAW, and Unrolled GANs. Discussing the relationship to those approaches would in my opinion add more depth to the paper.

---

### Meta-Review · Area_Chair1 · 2018-12-13
**interesting idea, evaluation lacking**

**Confidence:** 5
**Recommendation:** Reject

**Metareview:**

The paper presents a GAN-based generative model, where the generator consists of the base generator followed by several editors, each trained separately with its own discriminator. The reviewers found the idea interesting, but the evaluation insufficient. No rebuttal was provided.